# *Aruncus dioicus* var. *kamtschaticus* Extract Ameliorates Psoriasis-like Skin Inflammation via Akt/mTOR and JAK2/STAT3 Signaling Pathways in a Murine Model

**DOI:** 10.3390/nu14235094

**Published:** 2022-12-01

**Authors:** Banzragch Dorjsembe, Haneul Joo, Chuwon Nho, Jungyeob Ham, Jin-Chul Kim

**Affiliations:** 1Smart Farm Research Center, Korea Institute of Science and Technology, Gangneung 25451, Republic of Korea; 2Division of Bio-Medical Science and Technology, University of Science and Technology (UST), Daejeon 34113, Republic of Korea; 3Natural Product Research Center, Korea Institute of Science and Technology, Gangneung 25451, Republic of Korea

**Keywords:** *Aruncus dioicus* var. *kamtschaticus*, psoriasis, keratinocytes, imiquimod

## Abstract

Goat’s beard (*Aruncus dioicus* var. *kamtschaticus*) is a traditional medicinal plant, widely used in Chinese and Korean traditional medicine because of its anti-inflammatory, anti-oxidant, antimicrobial, and anti-cancer activity. However, its effect on skin inflammatory diseases like psoriasis is unknown. The aim of this study was to investigate the therapeutic potency of *A. dioicus* extract (ADE) in in vitro and in vivo psoriasis models. ADE treatment significantly attenuated skin inflammation and improved skin integrity in imiquimod-treated mice by suppressing keratinocyte hyperproliferation, inhibiting the infiltration of immune cells, and downregulating the expression of psoriatic markers. Further, ADE treatment suppressed protein kinase B/mammalian target of rapamycin (Akt/mTOR) and Janus kinase 2/signal transducers and activators of transcription 3 (JAK2/STAT3) signaling in HaCaT cells. Overall, the application of ADE relieves psoriasis-like skin inflammation possibly by regulating the Akt/mTOR and JAK2/STAT3 signaling pathways, making it an effective alternative for psoriasis therapy.

## 1. Introduction

Being one of the most common cutaneous inflammatory diseases, psoriasis affects 2–3% of the world’s population [1]. Its main features include hyperproliferative keratinocytes, scaly plaque, erythema, and infiltration of immune cells into the dermis. Initially, it was considered a skin inflammation but recent evidence indicates that it involves a complex communication between keratinocytes and immune cells in the skin [2,3].

Psoriasis progresses by means of signaling crosstalk via secreted cytokines. The complex crosstalk between immune cells and keratinocytes via cytokine signaling regulates onset of psoriasis and maintenance of inflammation in psoriatic skin. In detail, the flux of IFN-α from plasmacytoid dendritic cell populations, caused by external stimuli, skin damages and infection stimulate maturation of myeloid dendritic cells [4]. Once matured, the dendritic cells produce interleukin-6 (IL-6), IL-1β, and tumor necrosis factor-α (TNF-α), which induce the activation of naive T-helper cells (Th1, Th17, or Th22) [5,6]. Th cells migrate to the epidermis and produce interferon-γ (IFN-γ), IL-17a, and IL-22, thereby stimulating keratinocytes to express various chemokines, such as chemokine (C–C) motif ligand 20 (CCL20), chemokine (C–X–C) motif ligand 1 (CXCL1), CXCL2, and CXCL8, as well as antimicrobial proteins, such as β-defensin, LL-37, and proteins from the S100 family [7,8]. This vicious interplay between keratinocytes and immune cells promotes the infiltration of immune cells, such as dendritic cells, T cells, macrophages, and neutrophils, and the hyperproliferation of keratinocytes, aggravating the onset of psoriatic lesions [9,10]. The Janus kinase/signal transducer and activator of transcription (JAK/STAT) and the protein kinase B/mammalian target of rapamycin (Akt/mTOR) pathways are involved in the inactivation, differentiation, and proliferation of immune cells and keratinocytes in psoriatic lesions [11,12,13,14,15].

Most common therapy for psoriasis is topical steroid drugs, including corticosteroids and vitamin D analogues, widely used to alleviate mild psoriatic symptoms [16]. In moderate or severe case, immunosuppressive drugs, methotrexate and cyclosporine, and biological agents, which targets specific proteins, such as TNF-α inhibitor, etanercept, or IL-17A inhibitor, secukinumab are employed [17]. However, adverse effects like hypotension and lichenification and high commercial price limit availability of these drugs to every patient. Traditionally, many natural herbs and formulas have been used to treat inflammatory diseases like psoriasis and still practiced in world [18]. Due to their low toxicity and high abundance, natural products could be very attractive targets to develop novel therapeutic agents against psoriasis. *Aruncus dioicus* var. *kamtschaticus*, known as goat’s beard, is a perennial herb found in Russia, China, and Ulleung-do in Korea. It has been used as a food source and medicinal plant in traditional Korean and Chinese medicine [19,20]. Many studies have reported goat’s beard to possess antioxidant, anti-diabetic, antimicrobial, and neuroprotective activities [20,21,22]. However, its potency in treating skin inflammations like psoriasis remains obscure. The aim of this study was to evaluate the therapeutic potency of *Aruncus dioicus* extract (ADE) in an imiquimod (IMQ)-induced murine psoriatic model.

## 2. Materials and Methods

### 2.1. Plant Material Preparation and Extraction

Whole *A. dioicus* was purchased from the Hantaek Botanical Garden Foundation (Yongin, Gyeonggi-do, Republic of Korea) and identified by Dr. Jung Hwa Kang. The dried aerial parts of *A. dioicus* (2.89 kg) were extracted twice in 8 L of ethanol and evaporated under reduced pressure to yield the extract (120 g). The stock solution was prepared by dissolving in dimethyl sulfoxide (DMSO, Sigma Aldrich, St. Louis, MS, USA) and stored at −20 °C.

### 2.2. Animal Study

Male BALB/c mice (7 weeks old) were supplied by Orient Bio (Seongnam, Gyeonggi-do, Republic of Korea) and placed in pathogen-free animal facilities with free access to food and water. All experiments were approved by the Animal Care and Use Committee of the Korea Institute of Science and Technology (KIST-2018-091). Mice were randomly divided into five different groups (n = 5 per group): control group, NTC, IMQ-treated group, IMQ, 0.1 mg/mouse dexamethasone-treated group, Dex, Low dose (1 mg/mouse) ADE-treated group, ADE (L), and high dose (2 mg/mouse) ADE-treated group, ADE (H). The psoriasis-like model was established by daily application of 62.5 mg of IMQ Cream (Aldara™ Cream, Donga S&T, Seoul, Republic of Korea) for 1 week on shaved dorsal skin. Dex and ADE were dissolved in DMSO and diluted with PBS to final concentration. Mice were topically treated with 100 μL ADE or dexamethasone in PBS daily after the application of IMQ. Experimental procedures were evaluated by imaging analysis once in every other day and measurement of trans-epidermal water loss once in a 3 day. On the eighth day, all animals were sacrificed and the dorsal skin tissues were harvested. Psoriasis Area and Severity Index (PASI), adjusted according to the severity of scales, erythema, and infiltration on a scale of 0–4, was calculated to evaluate the skin condition. Correspondence scores were as follows: 0-none, 1-slight, 2-moderate, 3-marked, 4-very marked. The result was expressed as a cumulative PASI score [23].

### 2.3. Cell Culture and Treatment

The human immortalized keratinocyte cell line, HaCaT, was purchased from Addex Bio Technologies (AddexBio Technologies, San Diego, CA, USA). Cells were equilibrated in a 5% CO_2_ incubator at 37 °C and grown in high-glucose Dulbecco’s Modified Eagle’s Medium (HyClone Laboratories Inc, Grand Island, NY, USA). At 80% confluency, cells were starved in serum-free medium for 24 h, pretreated with 25, 50, or 100 µg/mL ADE for 1 h, and induced with 10 ng/mL TNF-α and 50 ng/mL IL-17a (Peprotech, Rocky Hill, NJ, USA) for 23 h before RNA extraction or for 30 min before protein extraction. Cells were incubated with wortmannin, a specific Akt inhibitor, and AG490, a JAK2 inhibitor, for 4 h before being stimulated for 20 h with a combination of IL-17a/TNF-α.

### 2.4. Cell Viability Assay

Cell viability was determined by the 3-(4,5-dimehtlythiazol-2-yl)-2,5-diphenyl-2H-tetrazolium bromide (MTT) assay. HaCaT cells were seeded in 96-well plates at a cell density of 1 × 10^4^ cells/well, incubated the next day in a fresh medium containing ADE (10, 25, 50, 100, or 200 µg/mL) for 24 h, washed with Dulbecco’s phosphate-buffered saline, and treated with MTT solution for 2 h. The formazan crystals were dissolved in 100 µL of DMSO and the absorbance was measured at 560 nm (reference wavelength of 650 nm) in an M1000 multiplate reader (Tecan, Mannedorf, Zurich, Switzerland).

### 2.5. Immunohistochemistry

Tissues were sliced from a paraffin block, dewaxed in xylene, hydrated using serially diluted ethanol, and stained with hematoxylin and eosin to investigate the skin structure and resident cells. Tris-EDTA buffer (1 mM, pH 9.0) was used for antigen unmasking. Tissues were blocked using 5% bovine serum albumin (Bioshop Canada Inc., Burlington, ON, Canada) in phosphate-buffered saline for 1 h, incubated overnight with primary antibodies against keratin 14 (K14, Biolegend, San Diego, CA, USA, 1:500), involucrin (IVL, Biolegend, San Diego, CA, USA, 1:500), and proliferating cell nuclear antigen (PCNA, Abcam, Cambridge, UK, 1:500) at 4 °C, treated with goat anti-rabbit (Abcam, Cambridge, UK) and goat anti-mouse (Abcam, Cambridge, UK) secondary antibodies, mounted using VECTASHIELD with DAPI (Vector Laboratories Inc, Burlingame, CA, USA), and observed under a Nikon Eclipse TE2000U microscope (Nikon Corporation, Tokyo, Japan). For staining with antibodies against CD3 (Abcam, Cambridge, UK, 1:200) and F4/80 (Abcam, Cambridge, UK), slides were stained using the VectaStain Elite ABC-HRP Kit (Vector Laboratories Inc., Burlingame, CA, USA, 1:200), according to the manufacturer’s guidelines. Color development was achieved using the DAB Substrate Kit (Peroxidase with nickel, Vector Laboratories Inc, Burlingame, CA, USA) and observed under an Olympus DP27 microscope (Olympus Korea, Seoul, Republic of Korea).

### 2.6. Quantitative Reverse Transcription-Polymerase Chain Reaction (RT–qPCR)

The easy-BLUE^TM^ Total RNA Extraction Kit (Intron Biotechnology, Seoul, Republic of Korea) and the QIAGEN RNeasy kit (QIAGEN, Valencia, CA, USA) were used to extract total RNA from mice tissues and cells, respectively. The RevertAid First Strand cDNA Synthesis Kit (ThermoFisher Scientific, Waltham, MA, USA) was used to reverse-transcribe RNA into cDNA and PCR amplification was achieved using PowerUp™ SYBR ™ Green PCR Master Mix (Applied Biosystems, Waltham, MA, USA), cDNA, and specific primer pairs in the QuantStudio 6™ Flex Real-Time PCR System (Applied Biosystems, Waltham, MA, USA), according to manufacturer’s instructions. The primers used and their sequences are listed in Table 1. Glyceraldehyde-3-phosphate dehydrogenase (GAPDH), a housekeeping gene, was used as an internal standard to normalize gene expression, which was calculated via the comparative Ct method [24].

### 2.7. Western Blot Analysis

Whole cell lysates were prepared and quantified using the bicinchoninic acid assay (Sigma Aldrich, St. Louis, MS, USA). Proteins from each sample (20 µg) were loaded on 10% sodium dodecyl sulfate–polyacrylamide gels, transferred to nitrocellulose membranes, and blocked in 5% bovine serum albumin for 1 h at room temperature. Membranes were incubated overnight at 4 °C with primary antibodies against phospho-JAK2 (Tyr1007/1008), phospho-STAT3 (Tyr705), phospho-mTOR (Ser2448), p-p70S6K (Thr389), phospho-Akt (Ser473), JAK2, Akt (pan), mTOR, p70S6K, STAT3, and GAPDH, diluted in 1x TBST solution containing 5% BSA with 1:1000 ratio, treated with mouse anti-rabbit IgG-HRP (Santa Cruz Biotechnology Inc., Dallas, Texas, USA), and analyzed using an iBright CL1000 system (Thermo Fisher Scientific, Waltham, MA, USA) after enhancement with SuperSignal™ West Femto Maximum Sensitivity Substrate (Thermo Fisher Scientific, Waltham, MA, USA). All primary antibodies were purchased from Cell Signaling Technologies (Danvers, MA, USA). The band intensity of Western blot was measured and quantified by Image J program.

### 2.8. Statistical Analysis

All experiments were performed independently at least thrice. Results are expressed as means ± standard deviations. The difference among groups were calculated by One-way Anova, followed by Bonferoni multiple comparision test, using IBM SPSS Statistics software (IBM, Armonk, NY, USA). *p* < 0.05 value was considered as statistically significant.

## 3. Results

### 3.1. Topical Application of ADE Reduced the Development of Psoriatic Phenotypes and Improved Skin Conditions

ADE or dexamethasone were applied daily for 1 week after psoriasis was induced using IMQ. Scaly patches, erythema, and acanthosis were observed in the IMQ group. Notably, treatment with ADE reversed these phenotypical changes and the ADE (H) group showed smoother skin with fewer scales and less erythema and infiltrates, comparable with the skin phenotype of the Dex group. Moreover, the total PASI score for the ADE (H) group reduced to a level similar to that of the Dex group. The ADE (L) group did not display significant phenotypical changes (Figure 1A).

Downward extended, thickened epidermis with damaged skin barrier and partial microabscesses were noticed in the skin tissues of the IMQ group. On the other hand, the administration of high dose of ADE considerably mitigated psoriatic symptoms and reduced epidermal thickness. The Dex group also showed skin with thinner and smoother epidermis (Figure 1B).

The uncontrolled hyperproliferation of keratinocytes dysregulates the normal skin barrier function, leading to a significantly thickened epidermis and increased epidermal water loss. These hyperproliferating keratinocytes in skin lesions produce higher amounts of K14 and IVL than in normal skin. ADE- or Dex-treated mice exhibited notably diminished production of K14 and IVL compared with IMQ-treated mice, as evidenced by immunohistochemical staining (Figure 1C,D).

PCNA is normally expressed in proliferating keratinocytes at the basal level. In IMQ-treated mice, the number of PCNA-expressing cells increased dramatically compared with that in control mice. ADE (L), ADE (H), and Dex groups showed significantly lower number of PCNA-positive cells compared with the IMQ group (Figure 1E).

### 3.2. ADE Treatment Attenuates the Infiltration of Immune Cells into Psoriatic Lesions by Suppressing the Secretion of Psoriatic Markers in Mouse Skin

Various immune cells, including T cells, macrophages, and neutrophils, migrate to psoriatic lesions in the epidermis; hence, preventing their infiltration is a therapeutic strategy for psoriasis. The effects of ADE on immune cell recruitment were evaluated by monitoring the populations of CD3^+^ T cells and F4/80^+^ macrophages using immunohistochemistry. Mouse tissues in the IMQ group showed a much higher number of CD3^+^ and F4/80^+^ cells compared with the control mice. However, a higher dose of ADE reduced the number of these cells in skin lesions to a level comparable to that in the Dex group (Figure 2A,B)

A unique panel of multiple inflammatory markers defines the symptoms and onset of psoriatic skin inflammation, so we investigated the effects of ADE treatment on the levels of these psoriatic markers using RT–qPCR. Compared with the control mice, the continuous application of IMQ to the dorsal skin of mice for 7 days significantly augmented the expression of interleukins (IL-1β and IL-23a), chemokines (CXCL1, CXCL2, CXCL10, and CCL20), and antimicrobial peptides (S100A8/9), all of which play a crucial role in the pathogenesis of psoriasis. In line with previous results, ADE treatment significantly suppressed the expression of these markers in psoriatic lesions. A similar pattern was observed in the Dex group, suggesting that ADE modulates the immune response in psoriatic skin lesions by downregulating inflammatory cytokines and other markers in the skin (Figure 3).

### 3.3. ADE Treatment Inhibits the Expression of Pro-Inflammatory Cytokines in HaCaT Cells

Keratinocytes produce several cytokines, which are crucial for the infiltration of inflammatory cells into psoriatic lesions [25]. ADE treatment did not cause any cytotoxicity, even at a dose of 200 µg/mL (Appendix A). In our experiment, HaCaT cells were treated with 25, 50, or 100 µg/mL of ADE. Psoriasis-related markers–interleukins (IL-1β and IL-23a), chemokines (CCL20, CXCL1, CXCL2, and CXCL10), and antimicrobial peptides (S100A8/9)–were upregulated after induction with IL-17a/TNF-α. ADE treatment attenuated the release of these markers from keratinocytes in a dose-dependent manner (Figure 4).

### 3.4. ADE Treatment Modulates JAK2/STAT3 and Akt/mTOR Signaling Pathways

The complex pathogenesis of psoriasis is regulated by the interplay between several signaling pathways, including JAK2/STAT3 and Akt/mTOR. We investigated the effects of ADE treatment on the activation of these signaling pathways using Western blot analysis. A combined induction of IL-17a/TNF-α activates these signaling cascades in keratinocytes. ADE significantly diminished the phosphorylation of JAK2/STAT3 and Akt/mTOR signaling molecules in a dose-dependent manner (Figure 5A,B). Furthermore, treatment with wortmannin, an Akt inhibitor, and AG490, a JAK2 inhibitor, induced similar inhibitory activities on the expression of pro-inflammatory cytokines (Figure 5C).

## 4. Discussion

The mechanisms governing the pathogenesis and onset of psoriasis remain unclear. Most reports have demonstrated that psoriasis is a chronic inflammatory autoimmune disease with a complex etiology. Topical agents like steroids are employed as first-line therapies against psoriasis. However, their use is limited by multiple adverse effects, such as allergic reactions, steroid addictions, and hypotension [26,27,28]. Therefore, finding a novel therapeutic agent to treat or prevent psoriasis symptoms is necessary. In this study, we used a murine model of psoriasis to determine the anti-inflammatory effect of ADE and a cellular model to elucidate its mechanism of action. We showed that ADE treatment suppressed the secretion of pro-inflammatory factors in keratinocytes via the JAK2/STAT3 and Akt/mTOR signaling pathways. Moreover, administration of ADE reduced psoriatic features, including parakeratosis, acanthosis, and hyperplasia, and prevented the infiltration of T cells and macrophages by downregulating the psoriasis hallmark genes in IMQ-treated murine skin.

One of the major phenotypic features in patients with psoriasis is hyperplasia or a thickened epidermis due to uncontrollably hyperproliferating keratinocytes and attenuated differentiation. In healthy skin, the delicate balance between systemic keratinocyte proliferation and differentiation maintains homeostasis for skin regeneration and barrier function. In the outer layers of the epidermis, keratinocytes differentiate into corneocytes, nucleus-absent cells, that act as physical barriers to protect against loss of humidity, external stress, and pathogen invasion [29].

For patients with psoriasis, this mechanism is usually dysregulated, leading to increased proliferation and decreased differentiation of keratinocytes. The Akt/mTOR signaling pathway is highly activated in skin lesions of such patients. The activation of phosphoinositide 3-kinase and Akt triggers mTOR to enhance keratinocyte proliferation while inhibiting their differentiation. Additionally, the upregulation of IL-17 and IL-22 activates mTOR in keratinocytes, indicating that the Akt/mTOR signaling pathway is a pivotal modulator of keratinocyte proliferation and differentiation in skin lesions. Further, inhibiting Akt signaling using its inhibitor, NVP-BEZ235, prevents IL-22-induced keratinocyte proliferation and the topical application of rapamycin and delphinidin ameliorates IMQ-induced skin inflammation and improves skin conditions in mice [14,30,31,32,33,34].

We demonstrated decreased phosphorylation of Akt, mTOR, and its downstream target, p70S6 kinase, in ADE-treated HaCaT cells. ADE-treated mouse skin displayed smoother and thinner epidermis with reduced K14 and PCNA expression. In addition, incubation with wortmannin, a phosphoinositide 3-kinase inhibitor, reduced the expression of pro-inflammatory cytokines, indicating that inhibition of Akt/mTOR signaling pathways might have regulatory effect on the production of inflammatory markers by keratinocytes in some extent. The JAK/STAT signaling cascade is also widely involved in many pathophysiological processes in psoriatic tissues, such as keratinocyte proliferation, production of inflammatory cytokines, and infiltration of immune cells. STAT3 signaling has been found to be increasingly activated in psoriatic lesions of patients or in IMQ-induced murine tissues. JAK signaling mainly acts as a downstream target of multiple cytokine receptors because it links cytokine signaling to their nuclear targets. JAK/STAT signaling plays an essential role in Th17 cell development and in the production of inflammatory cytokines, including IL-17a, IL-22, and IFN-γ [11,12,35,36,37,38,39].

In this study, ADE treatment reduced the phosphorylation of JAK2/STAT3 signaling molecules in HaCaT cells and downregulated the mRNA expression of psoriasis hallmark genes. Additionally, AG490, a JAK2 inhibitor, suppressed the production of psoriasis markers in cells. According to other reports, the blockade of JAK2 with ruxolitinib resulted in a positive therapeutic efficacy in psoriasis patients, and luteolin-7-glucoside and paeoniflorin compounds exerted their activity via JAK/STAT3 signaling in the IMQ-induced psoriasis-like murine model [40,41,42], suggesting prevention of psoriatic symptoms in IMQ-induced model by ADE treatment might be related to suppressive activity of ADE on activation of JAK/STAT signaling molecules.

In psoriatic skin, in response to cytokines, especially IL-17a or TNF-α from Th cells, keratinocytes produce several chemokines to attract immune cells into the epidermis. This cycle maintains local inflammation and upregulation of cytokines, thereby causing psoriatic symptoms and aggravating the inflammatory response in skin tissue [5,43,44]. We showed that ADE inhibited the production of multiple chemokines, such as CXCL1, CXCL2, CXCL10, and CCL20, as well as the antimicrobial peptides S100A8/9. Wortmannin and AG490 treatment also effected a similar reduction. All these results suggest that ADE exerts its anti-inflammatory activity in cellular and animal models by at least partially targeting the Akt/mTOR and JAK2/STAT3 signaling pathways. More detailed studies are necessary to reveal the exact mechanism.

## 5. Conclusions

In summary, ADE treatment diminished expressions of pro-inflammatory markers in in cellular inflammatory model via targeting Akt/mTOR and JAK2/STAT3 signaling pathways. Further, application of ADE notably mitigated psoriatic symptoms in IMQ-induced murine model. Thus, ADE might have beneficial effect on treating psoriasis.

## Figures and Tables

**Figure 1 nutrients-14-05094-f001:**
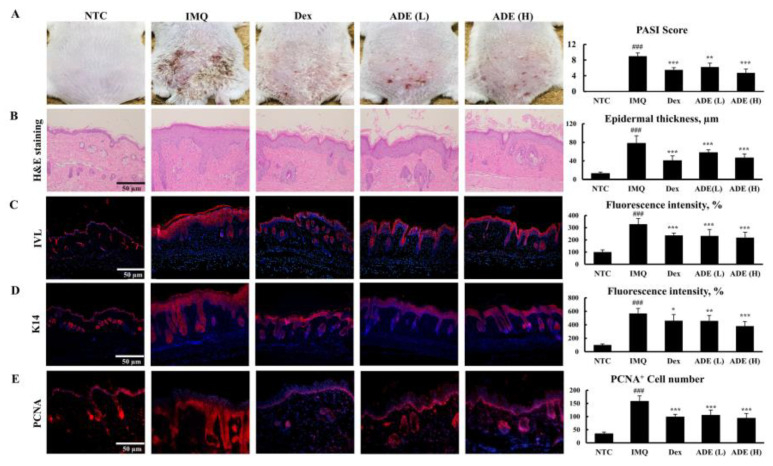
ADE treatment reduced IMQ-induced psoriasis symptoms. (**A**) Photo of dorsal skin tissue and cumulative PASI scores of the skin tissue. (**B**) Histopathological features were analyzed using H&E staining and epidermal thickness was measured (scale bar = 50 µm). Results are expressed as mean ± SD. ^###^
*p* < 0.001 vs. NTC group, *** *p* < 0.001 vs. IMQ group. The effect of ADE on the expression of K14 (**C**), IVL, (**D**), and PCNA (**E**) in psoriatic skin tissue was determined using IHC (scale bar = 100 µm, antibody dilution, 1:500) and was quantified. Results are expressed as mean ± SD. ^###^
*p* < 0.001 vs. NTC group, * *p* < 0.05, ** *p* < 0.01, and *** *p* < 0.001 vs. IMQ group. NTC: non-treated control, IMQ: imiquimod, Dex: dexamethasone, ADE (L): ADE (1 mg/mouse), ADE (H): ADE (2 mg/mouse). ADE, *Aruncus dioicus* extract; PASI, Psoriasis Area and Severity Index; H&E, hematoxylin & eosin; SD, standard deviation; K14, keratin 14; IVL, involucrin; PCNA, proliferating cell nuclear antigen; IHC, immunohistochemistry.

**Figure 2 nutrients-14-05094-f002:**
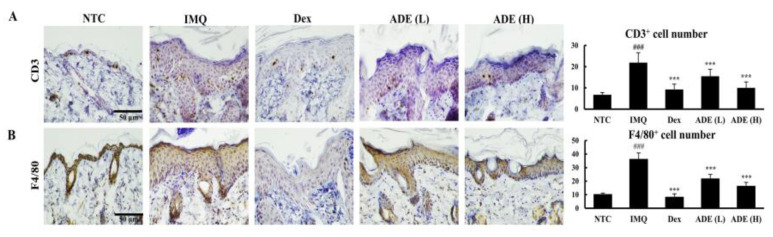
Topical administration of ADE attenuates the migration of immune cells to skin lesions in psoriatic mice. The infiltration of CD3^+^ T cells (**A**) and F4/80^+^ macrophages (**B**) was visualized by immunohistochemical staining and quantified (scale bar = 50 µm, antibody dilution, 1:200). Results are expressed as mean ± SD. ^###^
*p* < 0.001 vs. NTC group, *** *p* < 0.001 vs. IMQ group. NTC: non-treated control, IMQ: imiquimod, Dex: dexamethasone, ADE (L): ADE (1 mg/mouse), ADE (H): ADE (2 mg/mouse).

**Figure 3 nutrients-14-05094-f003:**
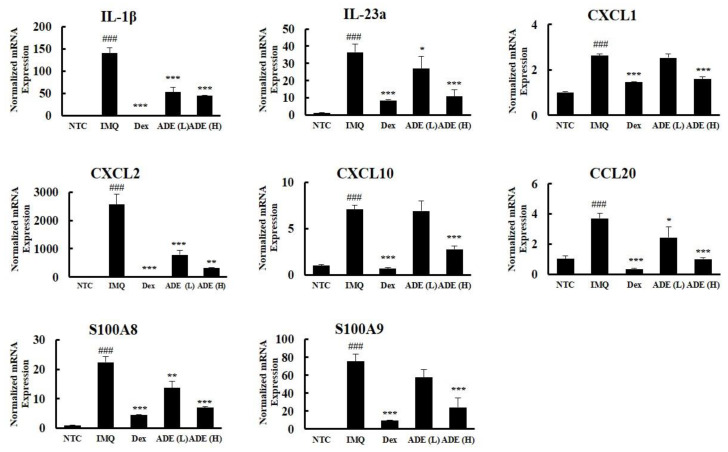
ADE treatment decreased the expression of psoriatic hallmark genes in skin lesions. The results of RT–qPCR analysis for inflammatory cytokines, chemokines, and antimicrobial peptides. Results are expressed as mean ± SD. ^###^
*p* < 0.001 vs. NTC group, * *p* < 0.05, ** *p* < 0.01, and *** *p* < 0.001 vs. IMQ group. NTC: Non-treat control, IMQ: Imiquimod, Dex: Dexamethasone, ADE (L): ADE (1 mg/mouse), ADE (H): ADE (2 mg/mouse). IL, interleukin; CXCL, chemokine (C–X–C) motif ligand; CCL, chemokine (C–C) ligand; RT–qPCR, quantitative reverse transcription-polymerase chain reaction.

**Figure 4 nutrients-14-05094-f004:**
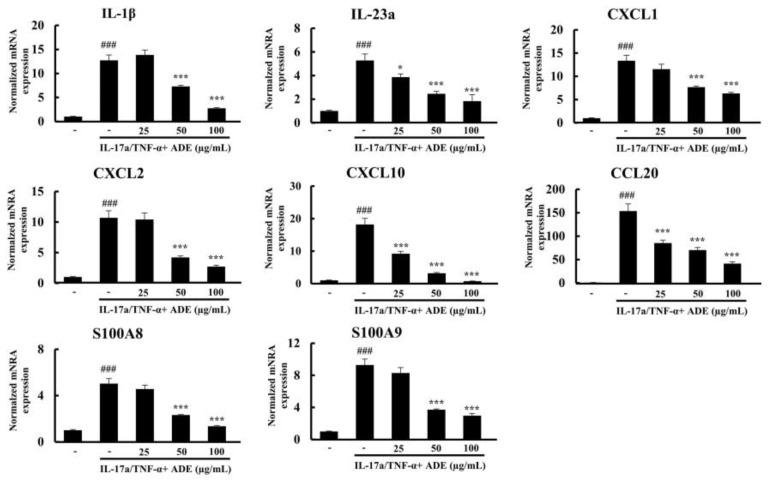
The mRNA expressions of IL−1β, IL−23a, CXCL1, CXCL2, CXCL10, CCL20, S100A8, and S100A9 in HaCaT cells, induced with IL−17a/TNF−α-induced for 24 h were measured by RT–qPCR. Results are expressed as mean ± SD. ^###^
*p* < 0.001 vs. NTC group, * *p* < 0.05 and *** *p* < 0.001 vs. IL−17a/TNF−α group. NTC: non-treated control, ADE: *A. dioicus* extract, TNF-α: tumor necrosis factor−α.

**Figure 5 nutrients-14-05094-f005:**
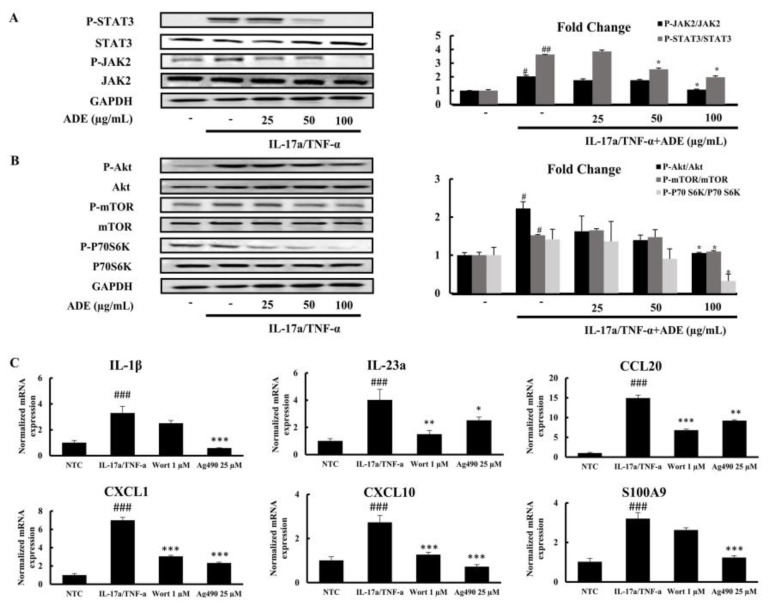
The effect of ADE on the activation of psoriasis−related signaling pathways, such as JAK2/STAT3 (**A**) and Akt/mTOR (**B**). The levels of different signaling molecules were determined using WB analysis in IL−17a/TNF−α-induced HaCaT cells. Relative fold change of phosphorylated proteins against total proteins was calculated using Image J software. Results are expressed as mean ± SD (n = 3). ^##^
*p* < 0.01 and ^#^
*p* < 0.05 vs. NTC group, * *p* < 0.05 vs. IL−17a/TNF−α group. NTC: non-treated control, ADE: *A. dioicus* extract. Roles of JAK2/STAT3 and Akt/mTOR pathways in IL−17a/TNF−α induced production of inflammatory cytokines in HaCaT cells (**C**). Cells were grown with specific inhibitors of Akt proteins (wortmannin) or JAK2 (AG490) for 4 h and induced with IL−17a/TNF−α for 20 h. The mRNA expressions of different cytokines were measured by qPCR. Results are expressed as mean ± SD. ^##^
*p* < 0.01 and ^###^
*p* < 0.001 vs. NTC group, * *p* < 0.05, ** *p* < 0.01, and *** *p* < 0.001 vs. IL−17a/TNF−α group. JAK/STAT, Janus kinase/signal transducers and activators of transcription; Akt/mTOR, protein kinase B/mammalian target of rapamycin.

**Table 1 nutrients-14-05094-t001:** List of primers.

Target Gene	Primer Sequence (5′-3′)	
	Forward	Reverse
*hIL-1β*	5′-TGAGCTCGCCAGTGAAATGA-3′	5′-AGATTCGTAGCTGGATGCCG-3′
*hIL-23A*	5′ACAGAAGCTCTGCACACTGG-3′	5′-GTTGTCCCTGAGTCCTTGGG-3′
*hCXCL1*	5′-ATTCACCCCAAGAACATCCA-3′	5′-TGGATTTGTCACTGTTCAGCA-3′
*hCXCL2*	5′-GCAGGGAATTCACCTCAAGA-3′	5′-TGGATTTGCCATTTTTCAGC-3′
*hCXCL10*	5′-TGCCATTCTGATTTGCTGCC-3′	5′-ATGCAGGTACAGCGTACAGTT-3′
*hCCL20*	5′-TGTCAGTGCTGCTACTCCAC-3′	5′-CCGTGTGAAGCCCACAATAA-3′
*hS100A8*	5′-AAGGGGAATTTCCATGCCGT-3″	5′-AGGACACTCGGTCTCTAGCA-3′
*hS100A9*	5′-CATGCTGATGGCGAGGCTAA-3′	5′-GCCTCGTGCATCTTCTCGTG-3′
*hGAPDH*	5′-GAAGGTGAAGGTCGGAGTC-3′	5′-GAAGATGGTGATGGGATTTC-3′
*mIL-1β*	5′-CAGGCAGGCAGTATCACTCA-3′	5′-AGGCCACAGGTATTTTGTCG-3′
*mIL-23A*	5′-CCATGGAGCAACTTCACACC-3′	5′-CTGGAGGCTTCGAAGGATCT-3′
*mTNF-α*	5′-TCTTCTCGAACCCCGAGTGA-3′	5′-CCTCTGATGGCACCACCAG-3′
*mCXCL1*	5′-GTCAGTGCCTGCAGACCAT-3′	5′-AACCAAGGGAGCTTCAGGG-3′
*mCXCL2*	5′-ACATCCAGAGCTTGAGTGTG-3′	5′-GCCTTGCCTTTGTTCAGTATCT-3′
*mCXCL10*	5′-TGAATCCGGAATCTAAGACCATCAA-3′	5′-AGGACTAGCCATCCACTGGGTAAAG-3′
*mCCL20*	5′-CGACTGTTGCCTCTCGTACA-3′	5′-AGCCCTTTTCACCCAGTTCT-3′
*mS100A8*	5′-ATGCCGTCTGAACTGGAGAA-3′	5′-TAGAGGGCATGGTGATTTCC-3′
*mS100A9*	5′-CAGCATAACCACCATCATCG-3′	5′-AAGGTTGCCAACTGTGCTTC-3′
*mGAPDH*	5′-CATGGCCTTCCGTGTTCCTA-3′	5′-ACTTGGCAGGTTTCTCCAGG-3′

## Data Availability

The data presented in this study are available upon request from the corresponding author.

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
