# Peer review of "Aruncus dioicus var. kamtschaticus Extract Ameliorates Psoriasis-like Skin Inflammation via Akt/mTOR and JAK2/STAT3 Signaling Pathways in a Murine Model"

_nutrients, 2022, doi:10.3390/nu14235094_

Round 1

Reviewer 1 Report

The authors present the animal (murine) study and cell model which aims to investigate the therapeutic potency of A. dioicus extract (ADE) in in vitro and in vivo psoriasis models. 

Comments:

1.

The statistical analysis and its method are used inappropriately in this study.

Student’s t-test for paired experiments is used in data analysis through all Figures.

In general, one-way analysis of variance (ANOVA) compares the means of two or more independent groups in order to determine whether there is statistical evidence that the associated group means are significantly different.

The Kruskal Wallis test is the non parametric alternative to the One Way ANOVA.

The one-way analysis of variance (ANOVA) was used for normally distributed variables, while the Kruskal-Wallis test was used for non-normally distributed variables.

Post-hoc tests are used at the second stage of the analysis of variance (ANOVA) or Kruskal Wallis test if the null hypothesis is rejected, e.g., a post-hoc tests were calculated by using Bonferroni multiple comparison test.

2.

Figure 5 (C) should be label. 

What does the Figure 5 (C) here mean?

Lines 273-275,

Furthermore, treatment with wortmannin, an Akt inhibitor, and AG490, a JAK2 inhibitor, induced similar inhibitory activities on the expression of pro-inflammatory cytokines (Fig. 5C).

It seems not ADE contain.

Author Response

Thank you for your comments on the manuscript. We modified the manuscript accordingly. 

Response to Reviewer 1 Comments

Point 1 The statistical analysis and its method are used inappropriately in this study.

Student’s t-test for paired experiments is used in data analysis through all Figures.

In general, one-way analysis of variance (ANOVA) compares the means of two or more independent groups in order to determine whether there is statistical evidence that the associated group means are significantly different.

The Kruskal Wallis test is the non parametric alternative to the One Way ANOVA.

The one-way analysis of variance (ANOVA) was used for normally distributed variables, while the Kruskal-Wallis test was used for non-normally distributed variables.

Post-hoc tests are used at the second stage of the analysis of variance (ANOVA) or Kruskal Wallis test if the null hypothesis is rejected, e.g., a post-hoc tests were calculated by using Bonferroni multiple comparison test.

Response 1:  Statistical significance was calculated with one-way anova methods, followed by post-hoc test of Bonferoni multiple comparision test.

Point 2: Figure 5 (C) should be label. 

What does the Figure 5 (C) here mean?

Response 2: Figure 5 (C) labeled in the figure legend as following “ Roles of JAK2/STAT3 and Akt/mTOR pathways in IL-17a/TNF-α induced production of inflammatory cytokines in HaCaT cells (C). Cells were grown with specific inhibitors of Akt proteins (wortmannin) or JAK2 (AG490) for 4 h and induced with IL-17a/TNF-α for 20 h. The mRNA expressions of different cytokines were measured by qPCR.”

Point 3: Lines 273-275,

Furthermore, treatment with wortmannin, an Akt inhibitor, and AG490, a JAK2 inhibitor, induced similar inhibitory activities on the expression of pro-inflammatory cytokines (Fig. 5C).

It seems not ADE contain.

Response 3: in Figure 5 (C), we have employed specific inhibitors of Akt proteins (wortmannin) or JAK2 (AG490) to investigate whether suppression of Akt/mTOR or JAK/STAT signaling affect on the transcription of pro-inflammatory genes and found out treatment with specific inhibitor decreased mRNA expression of several marker genes, IL-1β, IL-23a, CCL20, CXCL1 and S100A9 in absence of ADE.  Although still lacking conclusive evidences, since ADE treatment reduced phosphorylation of Akt/mTOR and JAK2/STAT3 signaling proteins along with upregulation of psoriasis hallmark genes and specific inhibitors showed similar activity, we predicted that ADE might exert its activity via possibly targeting Akt/mTOR and JAK2/STAT3 signaling pathway.

Reviewer 2 Report

The authors investigated the therapeutic efficacy of A. dioicus extract (ADE) in an in vivo and in vitro psoriasis model. They found that ADE treatment reduced skin inflammation and improved skin integrity in imiquimod-treated mice by suppressing keratinocyte hyperproliferation, inhibiting immune cell infiltration, and reducing the expression of psoriasis markers. As a conclusion, Authors suggest that application of ADE alleviates psoriasis-like skin inflammation, therefore it could be an effective alternative in the treatment for psoriasis.

Although the hypothesis is interesting, the design of the experiments and the evaluation of the results only allow speculations than confirming it. 

Comments to the Authors

(The first page of the paper is overlapping, enumeration and labeling are not in the right place that may be due to an inappropriate conversion into the pdf file.)

  1. Row 38: The sentence: “Psoriasis progresses by means of signaling crosstalk via secreted cytokines” is not clear. Please clarify these signaling crosstalks.
  2. Row 39 and 42: abbreviations and the used brackets should be checked (such as (IL)-6, abbreviation for interferon-γ
  3. Row 57: anti-oxidant should be replaced with antioxidant
  4. Row 68: dimethyl-sulfoxide abbreviation and company names are missing
  5. Row 77: Authors claimed: “(ADE (L))” I assume that the L stands for the low concentration while the (H) later is the higher concentration of the ADE. The authors need to clarify especially if they are introducing a new term/abbreviated expression.
  6. Row 80: “Mice were treated with ADE or dexamethasone daily after the application of IMQ.”  What does it mean treated? Did the authors use a food supplementation or cream as they did with the IMQ?
  7. Row 81: “Experimental procedures were evaluated by periodic imaging analysis and measurement of trans-epidermal water loss..” What on which day? As the author said later the mice experiment was terminated on day 8.
  8. Row 95: “At 80% confluency, cells were starved in serum-free medium for 24 h, pretreated with 25, 50, or 100 μg/mL ADE for 1 h, and induced with 10 ng/mL TNF-α and 50 ng/mL IL-17a (Peprotech, Rocky Hill, NJ, USA) for 23 h before RNA extraction or for 30 min before protein extraction.” What is the reason that the protein analysis was done in a relatively short period of time after the treatment while the RNA extraction was done after almost a day treatment?
  9. Row 99: “Cells were incubated with wortmannin, a specific Akt inhibitor, and AG490, a JAK2 inhibitor, for 4 h before being stimulated for 20 h with a combination of IL-17a/TNF-α.” Here, again the treatment parameters are different, however, if the authors are evaluating the inflammatory pathways of the current models, they really need to consider that TNF-alpha treatment can induce a severe response in 10 minutes even, thus what was the reason to pretreat the cells in such a long period of time?
  10. Row 102:  Were the MTT values normalized to protein content or cell numbers? 
  11. Row 112: missing antibody dilution ratios
  12. Row 150: bicinchoninic acid assay, missing its source
  13. Row 151: the sentence is confusing:Proteins from each sample (20 μg) were resolved”, what actually the resolved means?
  14. Row 152: missing percentages for the SDS-PAGE gels
  15. Row 153: missing information about the RT or cold incubation of the blocking solution
  16. Row 154: missing information about the antibody solutions 0.5 % or 5 % BSA in 1xTTBS? please clarify it
  17. Row 156: missing information about the antibody dilution ratios
  18. Figure 1: the figure was not precisely adjusted by the size, the pictures are rendered, and it needs to be corrected.
  19. Figure 1 Figure legend: antibody ratios are missing.
  20. Figure 2: see the comment of the figure 1
  21. Figure 3: see the comment of the figure 1
  22. Figure 3: The Y-axis titles needs to be changed to “Normalized mRNA expression on GAPDH”
  23. Figure 4 legends: the text is not clear on how long the treatment lasted, which is crucial if the authors are targeting the beneficial anti-inflammatory effects of the treatment.
  24. Figure 5: WB pictures are having a wide border, which makes the figure hard to read, even with magnification it is really hard to see the lanes in the case of the P JAK part. Hardly can see or even detect which band was considered as the phosphorylated one.
  25. Figure 5: the densitometric analysis was missing in the methods part, however there is a figure. The method needs to be clarified, which program was used for it, and how this fold change was made? Did the author first normalized on the GAPDH and then using the ratio-wise method? Also if the authors are using phosphorylated proteins, it is really important to know which amino acid residue is the one, especially in the case of the AKT and the mTOR ones, as we know from the literature, S 473 AKT is responsible for the cell survival, however, the T308 is for the protein synthesis, while the mTOR has 2 complexes, one for the protein synthesis and one for the cell survival processes. Figure legend did not contain how many repetition was made based on the calculations n=??
  26. Figure 5C: the same comment on Figure 3
  27. Row 327: The authors claimed: We demonstrated decreased phosphorylation of Akt, mTOR, and its downstream target, p70S6 kinase, in ADE-treated HaCaT cells. This phenomenon is supported by western blots.
  28. Row 330: „In addition, incubation with wortmannin, a phosphoinositide 3-kinase inhibitor, reduced the expression of pro-inflammatory cytokines, implicating the Akt/mTOR signaling pathway in the production of inflammatory markers by keratinocytes.” There is no data regarding the author’s statement. Using a strong statement like this needs to be supported by the expression of the PI3K pathway and other elements, like PIP3, PIP2, PDK1, and PDK2, otherwise, the sentence is far too speculative.
  29. Row 346: Consistent with our results, the blockade of JAK2 with ruxolitinib resulted in a positive therapeutic efficacy in psoriasis patients, and luteolin-7-glucoside and paeoniflorin compounds exerted their activity via JAK/STAT3 signaling in the IMQ-induced psoriasis-like murine model [36–38].” The authors did not show any of the inhibitory experiments, by which they could state that the results are consistent with the one from the literature.
  30. Row 355: The authors claimed that the treatment is decreasing the expression of the inflammatory molecules, however, they did not perform any of the protein expression targeting measurements. In this sense, the statement is false moreover speculative again. Based on the gene expression data it can not be concluded that the secreted-produced protein levels would be the same.

Author Response

Thank you for your comments on the manuscript. We modified the manuscript accordingly. 

Response to Reviewer 2 Comments

Point 1: Row 38: The sentence: “Psoriasis progresses by means of signaling crosstalk via secreted cytokines” is not clear. Please clarify these signaling crosstalks.

Response 1: We have explained about these crosstalk and added following sentences in to manuscript.

“The complex crosstalk between immune cells and keratinocytes via cytokine signaling regulates onset of psoriasis and maintenance of inflammation in psoriatic skin”.

Point 2: Abbreviations and the used brackets should be checked (such as (IL)-6, abbreviation for interferon-γ

Response 2: Edited abbreviation of interleukin-6 as following: “ interleukin-6 (IL-6), IL-1β…”

Added interferon gamma abbreviation as “IFN-γ” in the text.

Point 3: Row 57: anti-oxidant should be replaced with antioxidant.

Response 3:  Corrected word “anti-oxidant” to “antioxidant” in the manuscript.

Point 4: Row 68: dimethyl-sulfoxide abbreviation and company names are missing

Response 4: Added abbreviation of dimethyl-sulfoxide as following “ DMSO, Sigma Aldrich, St Louis, MS, USA)”

Point 5 : Row 77: Authors claimed: “(ADE (L))” I assume that the L stands for the low concentration while the (H) later is the higher concentration of the ADE. The authors need to clarify especially if they are introducing a new term/abbreviated expression

Response 5: Added explanation about low and high dose group as following: 

“Mice were randomly divided into five different groups (n = 5 per group): control group, NTC, IMQ-treated group, IMQ, 0.1 mg/mouse dexamethasone-treated group, Dex, Low dose (1 mg/mouse) ADE-treated group, ADE (L), and high dose (2 mg/mouse) ADE-treated group, ADE (H)”.

Point 6: Row 80: “Mice were treated with ADE or dexamethasone daily after the application of IMQ.”  What does it mean treated? Did the authors use a food supplementation or cream as they did with the IMQ?

Response 6: We have topically applied dexamethasone or ADE, diluted in PBS everyday after IMQ application and added explanation about animal treatment in manuscript as following

“Dex and ADE were dissolved in DMSO and diluted with PBS to final concentration. Mice were topically treated with 100 μL ADE or dexamethasone in PBS daily after the application of IMQ.”.

Point 7: Row 81: “Experimental procedures were evaluated by periodic imaging analysis and measurement of trans-epidermal water loss.” What on which day? As the author said later the mice experiment was terminated on day 8

Response 7: Imaging analysis performed once a every other day and TEWL was measured at once a 3 days. Added explanation into manuscript as following:

“Experimental procedures were evaluated by  imaging analysis once in every other day and measurement of trans-epidermal water loss once in a 3 days.”

Point 8: Row 95: “At 80% confluency, cells were starved in serum-free medium for 24 h, pretreated with 25, 50, or 100 μg/mL ADE for 1 h, and induced with 10 ng/mL TNF-α and 50 ng/mL IL-17a (Peprotech, Rocky Hill, NJ, USA) for 23 h before RNA extraction or for 30 min before protein extraction.” What is the reason that the protein analysis was done in a relatively short period of time after the treatment while the RNA extraction was done after almost a day treatment?

Response 8: Keratinocytes produce several types of inflammatory interleukins, chemokines and antimicrobial peptides in response to action of cytokines from other immune cells. Chiricozzi et al revealed that combination of IL-17a/TNF-α could induce increased expression of inflammatory cytokines in keratinocytes and 24 h incubation resulted in highest upregulation for most genes. Our unpublished data also indicate that IL-17a/TNF-α caused elevation of level of inflammatory cytokines in keratinocytes at 24 h, thus 24 h treatment was selected for RNA isolation.

Reference: Chiricozzi, A., et al., Integrative responses to IL-17 and TNF-alpha in human keratinocytes account for key inflammatory pathogenic circuits in psoriasis. J Invest Dermatol, 2011. 131(3): p. 677-87.

Point 9: Row 99: “Cells were incubated with wortmannin, a specific Akt inhibitor, and AG490, a JAK2 inhibitor, for 4 h before being stimulated for 20 h with a combination of IL-17a/TNF-α.” Here, again the treatment parameters are different, however, if the authors are evaluating the inflammatory pathways of the current models, they really need to consider that TNF-alpha treatment can induce a severe response in 10 minutes even, thus what was the reason to pretreat the cells in such a long period of time?

Response 9:

There are several reports that used wortmannin or AG490 to investigate their effect on expression of inflammatory cytokines such as TNF-α and CCL17 in monocyte or keratinocytes and those cells were pretreated with inhibitors 1-4 h and induced with 24-48h. Therefore, we have pretreated with HaCaT cells with wortmannin or AG490 for 4h.

References:

M.Ramirez et al, International Immunology, 11 (1999). 9,1479–1489, https://doi.org/10.1093/intimm/11.9.1479

S.M. Ju et al. Biochemical and Biophysical Research Communications 387 (2009) 115–120, https://doi.org/10.1016/j.bbrc.2009.06.137

K.Ohta et al, Molecular Medicine Reports. 16 (2017), 6850-6857, https://doi.org/10.3892/mmr.2017.7432

Point 10: Row 102:  Were the MTT values normalized to protein content or cell numbers? 

Response 10: MTT values were not normalized to protein content or cell numbers.

MTT values were calculated according to following methods:

  • Calculated average OD value of empty wells, containing only DMSO as blank
  • Subtracted average blank value from all measurement value at same plate.
  • Calculated an average for control wells, containing healthy cells with 100% viability
  • Calculated values of every sample as OD of sample/ Average OD for control*100% .

Point 11. Missing antibody dilution ratio.

Response 11: Specific dilution ratio for every antibody added in manuscript.

Point 12. Row 150: bicinchoninic acid assay, missing its source

Response 12: Added source of bicinchoninic acid as following:

“Whole cell lysates were prepared and quantified using the bicinchoninic acid assay (Sigma Aldrich, St Louis, MS, USA).”

Point 13. Row 151: the sentence is confusing:“Proteins from each sample (20 μg) were resolved”, what actually the resolved means?

Response 13: Changed word “resolved” to “loaded” to avoid confusion.

Point 14. Row 152: missing percentages for the SDS-PAGE gels

Response 14: Added gel percentage in manuscript as following:

“Proteins from each sample (20 µg) were loaded on 10% sodium dodecyl sulfate–polyacrylamide gels, transferred to nitrocellulose membranes, and blocked in 5% bovine serum albumin for 1 h at room temperature.”

Point 15. Row 153: missing information about the RT or cold incubation of the blocking solution.

Response 15: Added incubation condition of the blocking solution in manuscript as following:

“Proteins from each sample (20 µg) were loaded on 10% sodium dodecyl sulfate–polyacrylamide gels, transferred to nitrocellulose membranes, and blocked in 5% bovine serum albumin for 1 h at room temperature.”

Point 16. Row 154: missing information about the antibody solutions 0.5 % or 5 % BSA in 1xTTBS? please clarify it

Response 16: Added information about antibody solutions

“Membranes were incubated overnight at 4°C with primary antibodies … GAPDH, diluted in 1x TBST solution containing 5% BSA with 1:1000 ratio, treated …and analyzed … after enhancement ....”

Point 17. Row 156: missing information about the antibody dilution ratios

Response 17: Added information about antibody dilutions

“Membranes were incubated overnight at 4°C with primary antibodies … GAPDH, diluted in 1x TBST solution containing 5% BSA with 1:1000 ratio, treated …and analyzed … after enhancement ....”

Point 18. Figure 1: the figure was not precisely adjusted by the size, the pictures are rendered, and it needs to be corrected.

Response 18: Edited Fig.1 and inserted in manuscript.

Point 19. Figure 1 Figure legend: antibody ratios are missing

Response 19: Added antibody dilution ratio in Fig1 legend.

Point 20. See the comment of the figure 1

Response 20: Edited Fig.2 and inserted in manuscript. Added antibody dilution ratio in Fig2 legend.

Point 21. See the comment of the figure 1

Response 21: Edited Fig.3 and inserted in manuscript.

Point 22. The Y-axis titles needs to be changed to “Normalized mRNA expression on GAPDH”

Response 22: Changed Y-axis titles to “Normalized mRNA expression” and edited Figure 3, 4 and 5.

Point 23. Figure 4 legends: the text is not clear on how long the treatment lasted, which is crucial if the authors are targeting the beneficial anti-inflammatory effects of the treatment.

Response 23: Induction time added into Figure 4 legend.

“The mRNA expressions of IL-1β, IL-23a, CXCL1, CXCL2, CXCL10, CCL20, S100A8, and S100A9 in HaCaT cells, induced with IL-17a/TNF-α-induced for 23 h were measured by RT–qPCR.”

Point 24. Figure 5: WB pictures are having a wide border, which makes the figure hard to read, even with magnification it is really hard to see the lanes in the case of the P JAK part. Hardly can see or even detect which band was considered as the phosphorylated one.

Response 24: Wide border of WB pictures were edited and inserted to manuscript.

Point 25. Figure 5: the densitometric analysis was missing in the methods part, however there is a figure. The method needs to be clarified, which program was used for it, and how this fold change was made? Did the author first normalized on the GAPDH and then using the ratio-wise method? Also if the authors are using phosphorylated proteins, it is really important to know which amino acid residue is the one, especially in the case of the AKT and the mTOR ones, as we know from the literature, S 473 AKT is responsible for the cell survival, however, the T308 is for the protein synthesis, while the mTOR has 2 complexes, one for the protein synthesis and one for the cell survival processes.

Response 25: Densitometric analysis were analyzed by Image J program. To calculate fold changes, band densities of phosphorylated forms were normalized by total form and expressed ratio between sample and control group. Inserted to manuscript as following “ The band intensity of western blot was measured and quantified by Image J program”.

The information of phosphorylation site of proteins mentioned with antibody information as following:

“Membranes were incubated overnight at 4°C with primary antibodies against phospho-JAK2 (Tyr1007/1008), phospho-STAT3 (Tyr705), phospho-mTOR (Ser2448), p-p70S6K(Thr389), phospho-Akt(Ser473), JAK2, Akt (pan), mTOR, p70S6K, STAT3, and GAPDH, .., treated with mouse anti-rabbit IgG-HRP …, and analyzed.”

All experiments were repeated 3 times and added explanation in Figure 5 legends as following.

“The effect of ADE on the activation of psoriasis-related signaling pathways, such as JAK2/STAT3 (A) and Akt/mTOR (B). The levels of different signaling molecules were determined using WB analysis in IL-17a/TNF-α-induced HaCaT cells. Relative fold change of phosphorylated proteins against total proteins was calculated using Image J software. Results are expressed as mean ± SD (n=3). ##p < 0.01 and #p < 0.05 vs. NTC group, *p < 0.05 vs. IL-17a/TNF-α group.”

Point 26. The same comment on Figure 3.  

Response 26: Y-title axis were changed to “Normalized mRNA expression” and edited Figure 5.

Point 27 & 28. Row 327: The authors claimed: “We demonstrated decreased phosphorylation of Akt, mTOR, and its downstream target, p70S6 kinase, in ADE-treated HaCaT cells.” This phenomenon is supported by western blots.

Row 330: In addition, incubation with wortmannin, a phosphoinositide 3-kinase inhibitor, reduced the expression of pro-inflammatory cytokines, implicating the Akt/mTOR signaling pathway in the production of inflammatory markers by keratinocytes.” There is no data regarding the author’s statement. Using a strong statement like this needs to be supported by the expression of the PI3K pathway and other elements, like PIP3, PIP2, PDK1, and PDK2, otherwise, the sentence is far too speculative.

Response 27&28: We corrected statement as following: “We demonstrated decreased phosphorylation of Akt, mTOR, and its downstream target, p70S6 kinase, in ADE-treated HaCaT cells. … In addition, incubation with wortmannin, a phosphoinositide 3-kinase inhibitor, reduced the expression of pro-inflammatory cytokines, indicating that inhibition of Akt/mTOR signaling pathways might have regulatory effect on the production of inflammatory markers by keratinocytes in some extent.”

Point 29. Row 346: “Consistent with our results, the blockade of JAK2 with ruxolitinib resulted in a positive therapeutic efficacy in psoriasis patients, and luteolin-7-glucoside and paeoniflorin compounds exerted their activity via JAK/STAT3 signaling in the IMQ-induced psoriasis-like murine model [36–38].” The authors did not show any of the inhibitory experiments, by which they could state that the results are consistent with the one from the literature.

Response 29: In this study, ADE sample inhibited phosphorylation of JAK2/STAT3 protein and inhibition of JAK2 signaling with specific inhibitor, AG490 resulted in downregulation of several inflammatory cytokines in IL-17a/TNF-α-induced keratinocyte model. This suggests that JAK2/STAT3 signaling pathways involves in expression of several psoriasis hallmark genes in keratinocytes in certain extent. In addition, several reports found out that targeting Jak/Stat signaling prevents psoriatic symptoms in animal models. Therefore, we can speculate that inhibition of jak/stat with ADE treatment might have potential effect on reducing psoriasis-like skin inflammation.

However, there is no conclusive evidences of ADE targeting on JAK/STAT signaling in animal model, we changed sentence into

“According to other reports, the blockade of JAK2 with ruxolitinib resulted in a positive therapeutic efficacy in psoriasis patients, and luteolin-7-glucoside and paeoniflorin compounds exerted their activity via JAK/STAT3 signaling in the IMQ-induced psoriasis-like murine model [40-42], suggesting prevention of psoriatic symptoms in IMQ-induced model by ADE treatment might be related to suppressive activity of ADE on activation of JAK/STAT signaling molecules. ”

Point 30. Row 355: The authors claimed that the treatment is decreasing the expression of the inflammatory molecules, however, they did not perform any of the protein expression targeting measurements. In this sense, the statement is false moreover speculative again. Based on the gene expression data it can not be concluded that the secreted-produced protein levels would be the same

Response 30: Originally, we intended to state that ADE treatment is decreasing mRNA expression of the psoriasis hallmark genes. Thus to avoid confusion, changed sentence into as following

“ In this study, ADE treatment reduced the phosphorylation of JAK2/STAT3 signaling molecules in HaCaT cells and downregulated the mRNA expression of psoriasis hallmark genes.”

Round 2

Reviewer 1 Report

No further comment  

Reviewer 2 Report

none